# Extension of the HEMRM—Full Harmonization of the Electricity Supply System

**Zoran Marinšek [1],\*, Sašo Brus [2] and Gerhard Meindl [3]**

[1] OFFSET Energy d.o.o.; Competence Center for Advanced Control Technologies (KC STV), Tehnološki park 18, 1000 Ljubljana, Slovenia

[2] OFFSET Energy d.o.o., Tehnološki park 19, 1000 Ljubljana, Slovenia; saso.brus@offsetenergy.eu

[3] SWW Wunsiedel GmbH; Es-geht!-Energiesysteme GmbH, Hauptstrasse 117, 10827 Berlin, Germany; gerhard.meindl@es-geht.gmbh

\* Correspondence: zoran.marinsek@offsetenergy.eu

**Abstract:** The current formal common denominator of the electricity supply system in Europe has been the Harmonized Electricity Market Role Model (HEMRM) set up by ENTSO-E, ebIX, and EFET at the turn of the millennium; it introduced the concept of de-coupling and the vertical structuring of the system into the previously vertically integrated system. Since then, within demonstration projects, the system has been undergoing further changes in a controlled environment, generating bottom-up energy, caused by new technologies, business models, and new players, and extending the concept of the system to the level of energy communities and prosumers. Therefore, this paper proposes a coherent approach to the extension of HEMRM to the lowest levels in both the grid and market segments—full harmonization. This entails further structuring of both segments downwards and applying the principles of vertically nested subsystems—a system of systems approach—to a unit functional level of the electricity system, which can be the prosumer itself. At the lowest levels, the de-coupled system becomes coupled; additionally, it cross-sects with other energy vectors. Complete harmonization reduces the number of system and market segments and represents system standardization, leading to both subsystem and system-wide optimization. Prerequisites for it include the automated trading of flexibilities by the prosumers and implicit trading of energy transfer capacities along the distribution grids. The energy reservoirs, implicit and explicit, short-term, and long-term, play a vital role in techno-economic balancing.

**Keywords:** HEMRM; de-coupling; prosumer; energy community; harmonization; vertically nested systems; sector coupling; flexibility

## 1. Structure of the Electricity Market System in Europe

The current formal common denominator of the electricity supply system in Europe has been the Harmonized Electricity Market Role Model (HEMRM), which introduced the concept of de-coupling and vertical structuring into the previously vertically integrated and uncoupled system.

### 1.1. Harmonized Electricity Market Model in Europe

The original HEMRM is the result of cooperation between three major stakeholders in the European market, ETSO—European Transmission System Operators (presently ENTSO-E), ebIX—European Forum for Energy Business Information Exchange, and EFET—the European Federation of Energy Traders, at the turn of the millennium. The work of ETSO started in 2001, with the other two partners joining subsequently, and has continued with several milestones. In 2009 [1], the model was harmonized on lower levels but not on top level(s); there were several updates and modifications after that.

However, in this paper, we shall mostly refer to the 2009 version, due to its most complete view of the concept. In later stages, the evolution of the model has occurred,

motivated by new directives and interactions with other stakeholders from major industry players and other active configurations of stakeholders, e.g., the European Electricity Grid Initiative. The complete list of versions to date is given in [2].

The model is based on the concept of bringing the market to electricity generation and supply systems, i.e., to treat energy as a marketable product.

The model has been termed the "Harmonized Electricity Market Role Model", but it comprehensively addresses, on the one hand, the organization and structuring of players and, on the other hand, the processes which (should) constitute the electricity market and those which are necessary to assure the operational capability of the electricity grids in these new circumstances.

Prompted by European directives, the evolution of the Harmonized role model has been accompanied by gradual diffusion into national role models and national regulations covering the organization of national electricity markets.

In the original version of the model (2009), there are two complementary views:

- The basic (ETSO) view: The Harmonized Electricity Market Role Model version 2009-01 [1];
- The process (ebIX) view: UMM 2 Business Requirements View for the structuring of the European energy market, carried on as the ebIX Domain Model [2].

In the basic (ETSO) view, the model is represented by roles, domains, their inter-relations, and interactions, as shown in Figure 1, copied from [1]. The different types of interactions between different roles are symbolically depicted by exchanged business messages—arrow lines of different colours.

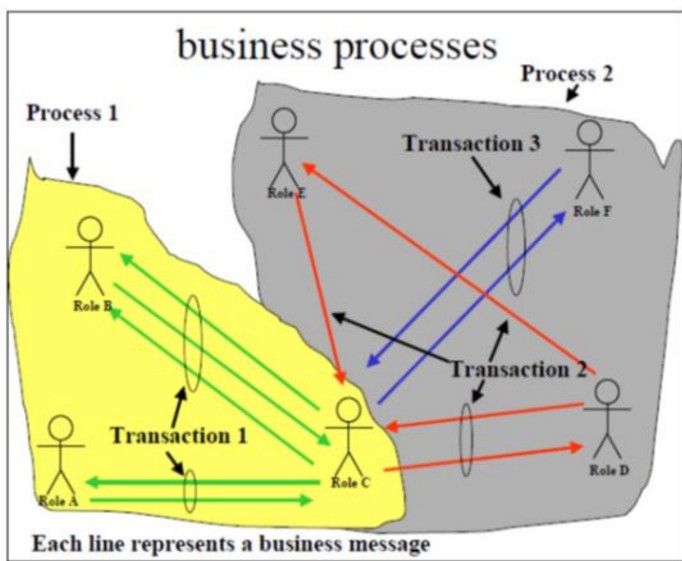

**Figure 1.** Business processes, transactions, and messages [1].

While the basic representation of the Harmonized role model focuses on the organizational aspects of the market (roles and domains), the ebIX view focuses on structuring the processes of the electricity market. The basic view from this viewpoint is depicted in Figure 2, copied from [2]. This view is not presented in later versions of HEMRM.

The structuring of the roles and domains (systems) provides for the separation of commercial activities in production, trading, and the consumption of energy from the activities of maintaining the grid, which must be available to users on equal terms and is therefore a public-function type of activity.

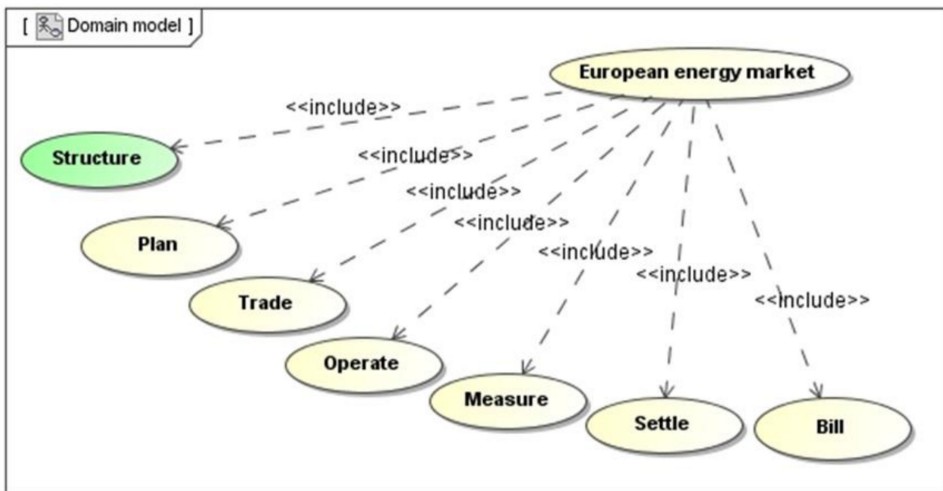

**Figure 2.** The phases of the European energy market in the ebIX model [2].

*1.2. Observed Characteristics of the HEMRM*

To fully address the characteristics of the electrical energy market system, the viewpoint relevant to the task of managing and controlling the electrical energy market system—the viewpoint of the system engineer—is selected. There are three main constituents to this approach:

- The structure of the system itself and its environment;
- The roles of the entities that constitute the system;
- The processes that occur in these roles.

1.2.1. Structuring of the Electricity Market System

By observing the original version of the Harmonized Electricity Market Role model and related HEMRM 2009 [1], and later versions until 2018 [3], documents, it can be seen that the electrical energy market system can be vertically and horizontally decomposed–structured.

The main line of vertical decomposition follows the concept of nested fractal subsystems, with the subsystems on each nested level having essentially the same characteristics as their parental system on the next higher level while being consistent with the level of decomposition. This fractal-like characteristic applies to both the roles and to the processes carried out by the roles and their interactions. We shall refer to these types of subsystems as *primary subsystems* and the processes that occur in them as *primary processes*. They are also frequently referred to as "cellular systems".

Horizontal decomposition refers principally to a number of fractal-like subsystems, which exist in parallel on the same level.

In addition to this structure of primary subsystems, there is one subsystem that is not vertically structured in levels, or, rather, its structure does not completely follow the structure of the vertically and horizontally decomposed primary subsystems. It has the specific functions of *joint and supportive processes* necessary for the operation of the electricity market, which maintain the electricity grid, in a technical and business sense. In the context of the management of processes in the primary subsystems, it can be considered as the "environment" of the primary subsystems on each vertical level, defining the boundary operating conditions of these systems. We designate this type of subsystem as a *structural subsystem*, and the processes within the structural subsystem as *joint and supportive processes*.

This characteristic was first observed and worked upon in the MIRABEL project [4] (together with FlexOffer protocol [5] which provides for the high-level parametric "standardization" of flexibility trading) and in parallel in several national projects in Denmark and Slovenia as well; it has been further developed and expanded in the GOFLEX project [6,7] (pp. 47–52).

Schematically and conceptually, this vertical and horizontal decomposition of the electricity market is illustrated in Figure 3. On each vertical level, several similar primary subsystems exist, with one modeled structural subsystem, depicted as hexagonal, containing the joint and supportive processes that interact with the processes in all of the primary subsystems on this vertical level. In the graphic, four levels of vertical decomposition are sketched.

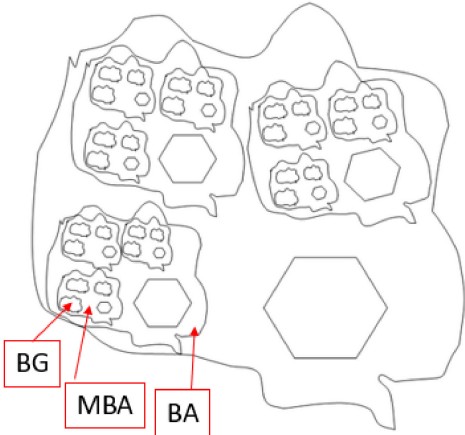

**Figure 3.** Schematic representation of the vertical and horizontal decomposition of the electrical grid system into nested subsystems.

The vertical structure of the primary subsystems in the state of the Harmonized Electricity Market Role Model (2009 issuance) consists of four levels of primary subsystems of the Electricity Market system:

- 1st level: Balance Group (BG);
- 2nd level: Market Balance Area (MBA) (*);
- 3rd level: Market Area (Local Market Area) (BA) (*);
- (4th level: European) (extrapolated).

(*) In later versions of the HEMRM (after 2018), this structure was modified; thus, MBA was changed to Scheduling Area (SA) and BA was deprecated [3]. In this paper, we will keep referring to the MBA.

### 1.2.2. Roles and Processes

The process of the electricity market and grid system consists of energy production, transmission (flow of energy), consumption, and trading. The primary process in the electricity market system is the trading of energy.

#### Primary Process

The primary process has to be broken down into *unit processes* according to the need of the technology employed or available but also consistently with the boundary conditions of the Harmonized model.

The trading process is based on closed contracts using flex-offers [5] issued by prosumers. It includes the following major unit processes: planning (production/consumption), offering (flex-offer), aggregation, scheduling, auctioning, assigning (contracting), de-aggregation, executing (production/consumption), and settlement.

The generic scope of unit processes and the unit processes implemented in the GOFLEX demonstration cases can be seen in its requirements analysis [6].

The proper structuring of the overall primary process into unit processes, and showing that these unit processes occur at different levels of the vertically decomposed system, makes it possible to use trading technology on different levels of the system—i.e., scale it simply to the system where it is used so there is no (appreciable) difference for the unit process if different roles are involved—if the unit processes are properly defined (e.g., the

negotiation process is the negotiation process whether it is performed between consumer and BRP or between BRP and BRP, etc.).

For this reason, the above list of unit processes is tentative in scope and generic in formulation. The final scope of the list and the specific nominations of unit processes are tailored to individual use cases, as seen in Section 4 of this document. The use cases also provide a check that the final unit processes are exhaustive.

Joint and Supportive Processes

Similarly, the processes in joint and supportive subsystems are structured into unit processes. These include measuring, forecasting (production, consumption, losses, and congestion), operating the LV/MV distribution system, operating the HV transmission system, and others.

The joint and supportive processes, as such, are of interest in the extent to which the primary processes interact with them. This means that we limit ourselves to identifying the interacting roles, without looking into the processes carried out by the role, and we only consider immediately adjacent roles.

An exception to this approach is the new DSO and sub-DSO roles (cf. Section 2.3), which participate in the new use cases for the local balancing of the local grid. These new roles and use cases greatly enhance the interaction between the front-end (primary processes) and the back-end (joint and supportive processes) segments of the electricity system, and, as users of energy flexibility, contribute significantly to driving the use of local demand side management for energy balancing when there are increased shares of renewables on the grid.

Roles

The roles inside a primary subsystem carry out the primary process, are structured into unit processes, and interact with the processes in the environment of the subsystem carried out by the neighboring roles.

In HEMRM, of particular importance is the subsystem Balance Group, as it contains all parties connected to the grid. It is delineated in the cut-out segment of the HEMRM in Figure 4, taken from [4].

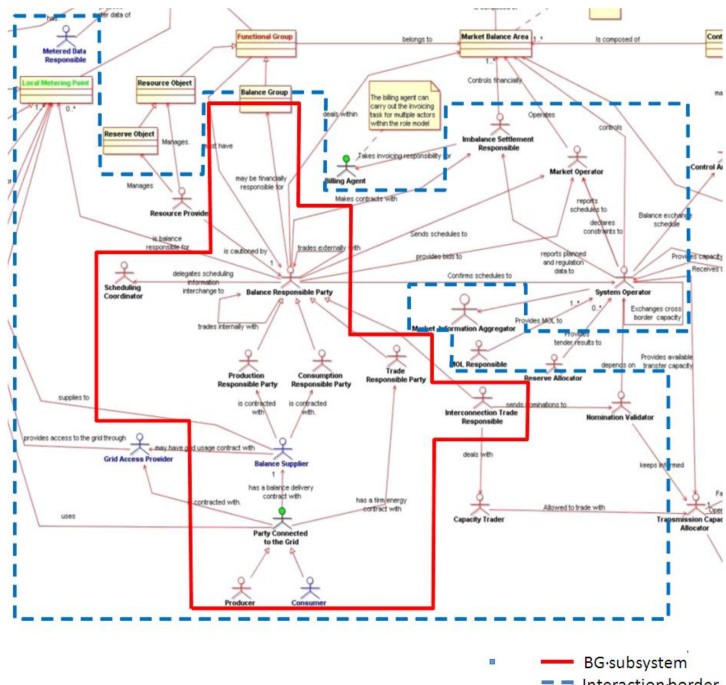

**Figure 4.** Delineation of the Balance Group subsystem and its interaction border with the surrounding part of the Electricity Market of the Harmonized market model, version 2009 [4].

The roles inside the BG carry out the primary process, structured into unit processes, and interact with the processes in the environment of the BG carried out by the neighboring roles. The dashed line represents the interaction border.

## 2. Extension of HEMRM—Full Harmonization of the Electricity Market System

The fractal-like characteristic observed in the original HEMRM applies to both the roles and to the processes carried out by the roles and their interactions.

This opens the doorway to a new market design with a number of use cases with "essentially the same characteristics", i.e., they are scalable to different levels and represent a path to "systemic standardization" in electricity market design. This has several important impacts: the simplification of interactions/coupling between subsystems that are both on the same level (peer-to-peer) as well as on connected vertical levels, which leads to a greater security of supply; enables both intra-system optimization and optimization of complete MBA system; reduces the necessary amount of specific regulations; and provides for scalability in technologies.

As stated, this approach has been further developed and expanded in the GOFLEX project as GOFLEX Roles and the process model [7], and has been further applied to types of systems on geographical islands in project GIFT [8] and project FEVER [9]. The model has been fully described in GOFLEX Deliverable D6.2 [7].

The GOFLEX Roles and process model involve further harmonization of the Electricity Market system, i.e., extension of the principles of the Harmonized Electricity Market Role Model in both markets and in grid segments to the lowest system level; in the market level to the sub-balance group, micro-grid, energy community and other emerging types of local subsystems; in the grid level to DSO; and to the sub-DSO level where applicable.

The approach has been presented and debated in several concept-shaping events: e.g., the E-DSO workshop in April 2019 [10], the GOFLEX Workshop on EUW19 (together with FlexOffer concept) in October 2019 [11], the BRIDGE Assembly in February 2020 [12,13], and the CIRED Berlin Workshop in September 2020 [14], The following four Sections 2.1–2.4 represent a summary description of the concept.

### 2.1. MBA Subsystem and Roles (Potentially) Involved in Processes for Trading Energy

In the MBA subsystem within the Harmonized Electricity role model, the roles that are potentially involved in trading energy, and their relations, are depicted in the following schematic (Figure 5):

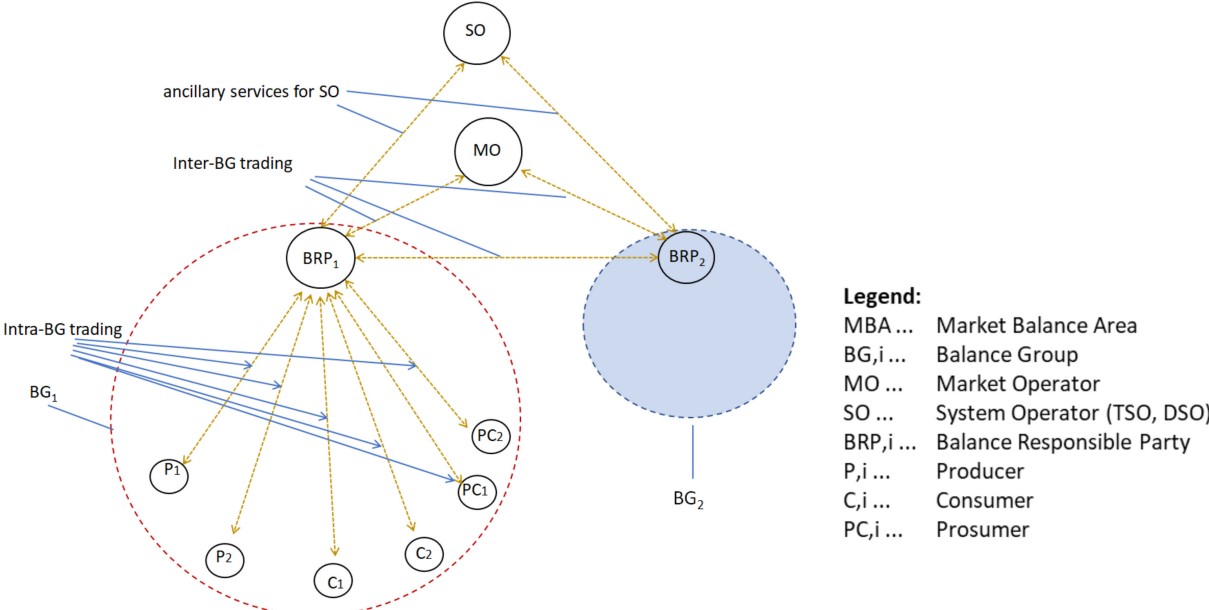

**Figure 5.** MBA with BG systems with the roles that participate in energy trading.

The case presented is an MBA with two BGs; the energy trading processes are shown with dashed two-arrow lines:

- Intra-BG trading between parties connected to the grid and BRP,1 in BG1;
- Inter-BG trading in MBA between $BRP_1$ and $BRP_2$, either through the Market Operator or directly';
- Trading (supplying) ancillary services for the tertiary reserves of SO.

In the original Harmonized model, there is no structuring of SO; it contains the TSO and all DSOs in its structure. This schematic is an example of all possible use cases for energy trading within the Harmonized model. The potentially involved roles are listed in the legend.

### 2.2. Further Vertical Structuring of Electricity Market System (Primary Subsystems) Downwards

The vertical decomposition of the primary system into the vertically nested primary subsystems identified in project GOFLEX [7] encompasses the following types of subsystems:

- Sub-balance group (or Balance Sub-group): is completely vertically nested in its parental subsystem Balance Group.
- Local Community Micro-grid: consists of members (roles) on a territory of a segment of the distribution grid that fulfill the condition that they could operate separately from the grid. It includes the lowest voltage level of the electrical grid and has a limited number of connections with the rest of the grid (cf. Figure 6).
- Local Energy Community: is a similar subsystem to the local community microgrid, with the addition that it includes all types of energy supply to the community: electricity, thermal energy, etc. These supply systems interact on the community level and provide constituents for overall energy efficiency.
- Prosumer: In this concept and approach, the prosumer is the lowest-level subsystem in the electricity system.

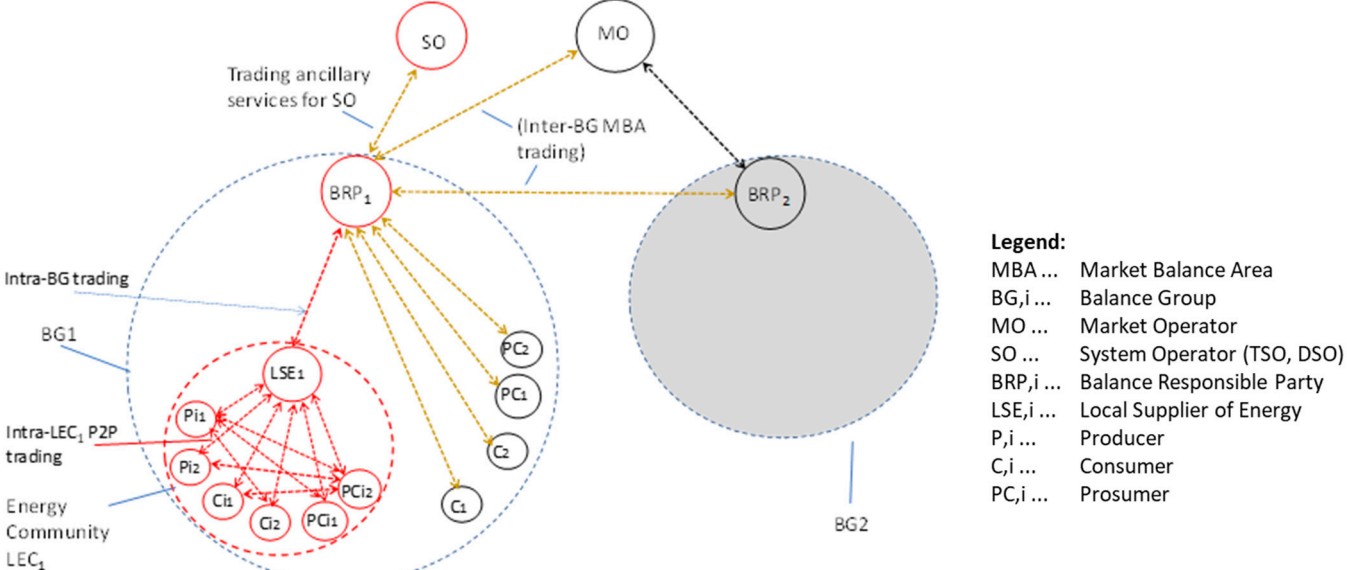

**Figure 6.** Connected local energy community system with roles that participate in energy flexibility trading internally and within the Balance Group.

In such a structure, energy trading takes place on each level, and each system tries to optimize its operation techno-economically, in terms of the efficient use of energy and balancing of supply and demand. In such a system, the prosumer plays the central role, and the flexibility trading—demand response of prosumers—is the key category to engage their active role.

The new primary subsystems are or tend to become local. In this context, the Local energy community is indicative and can be considered representative of their role and potential in energy supply.

It is important to note that, due to the concept of trading, the systems tend to organize themselves to operate optimally bottom up: Prosumer, Local Energy Community, and Balance Group.

With energy trading on the prosumer level, the self-supply on the level of the prosumer and the local level is carried out as part of its techno-economical optimization and to the extent that it resides within the objectives of

- The cost-effective efficient use of energy;
- Cost-effective energy balancing.

In addition to energy trading by the prosumers, another necessary ingredient for techno-economical optimization is the dynamic price of energy flexibility based on local conditions on the grid and stemming from avoided costs when using the energy flexibilities, as opposed to competitive investments into "peaker" production capacities.

### 2.3. Further Harmonization of the System in the Segment of the Electricity Grid

The Harmonized model, with its monolithic structure of the structural system for joint and supportive processes, has increasingly proved inadequate at providing a balanced counterpart to the intense evolution of technologies and use cases, resulting in business models on the front end of the electricity system that are necessary to cope efficiently (technology wise and financially) with increasing shares of RES.

As observed and formulated by GOFLEX [7], similarly to the primary system, the electricity grid system within the MBA can be harmonized into vertically nested structural subsystems:

- MBA (TSO territory);
- DSO territory;
- Sub-DSO territory.

This harmonized structure of the grid segment is shown in Figure 7.

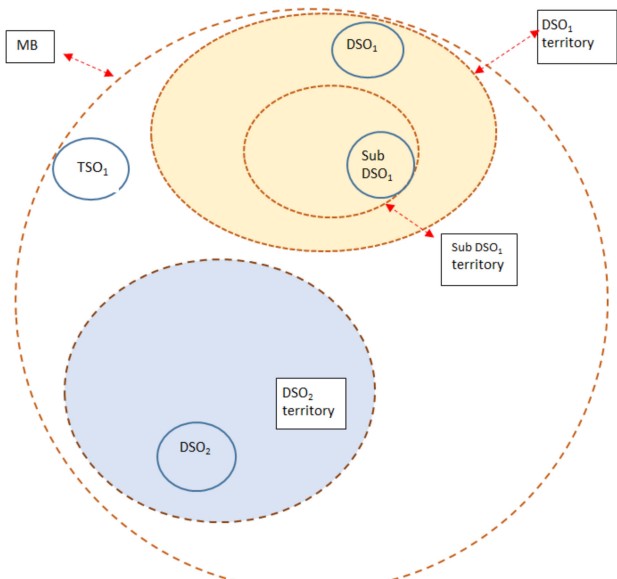

**Figure 7.** Vertical structuring of the grid segment of the system: TSO territory into DSO territories and into sub-DSO territories.

Consequently,

1. The DSO system becomes a "cellular" subsystem of the TSO system and DSO becomes responsible for its distribution grid as a vertically nested system into TSO—its parental system (="cellular" DSO).

   - It is responsible for all functions controlling the grid on its level, including the responsibility of balancing the local energy flows on the grid. This entails a new business model for DSO (cf. Section 2.6.3).

2. The vertical structuring involves or can involve several levels, following the voltage level structure of the distribution grid.

   - DSO subsystem: subsystem consisting of the grid on a high distribution level (110 kV);
   - Sub-DSO, level 1: subsystem consisting of the grid on the middle distribution level (20 kV);
   - Sub-DSO, level 2: subsystem consisting of the grid on a lower distribution level (0.4 kV).

3. Concrete vertical structuring is performed in a way to mirror the structuring of primary subsystems on a local level, local community micro-grid, and/or local energy community. In this way, they can provide an optimal cross-section (cf. Section 2.3).

*2.4. Coherent Structuring—Cross-Sections between Both Segments of the Electricity System*

The Balance Group system is not territorial, i.e., its roles can be located anywhere in the MBA. The same applies to subsystem 1, in Section 2.1, the Sub-Balance group system. On this principle, the Sub-Balance Group can be further vertically decomposed.

The characteristic of subsystems 2 and 3 (and 4), listed in Section 2.1, the local community micro-grid, and the Local Energy Community (and Prosumer), is that while being subsystems of the Balance Group, they are also territorial, i.e., their roles are located on a certain compact part of the territory within BG.

Thus, systemically, they are vertically nested only if they are also in the cross-section of the Balance Group and the applicable part of the structural system. Conversely, being in the cross-section between the two is a necessary condition for positioning either of these two systems as vertically nested subsystems in the Balance Group.

Consequently, the structuring in both segments (market and grid) of the electricity market system must be performed "coherently" so as to provide cross-sections between them to:

- Enable engaging the flexibilities locally;
- Enable the provisioning of all types of services for the DSO and by the DSO.

The concept is shown in Figure 8. The objective is to maximize local problem-solving on each nested subsystem level.

*2.5. Summary: Systemic Characteristic of Harmonized Electricity MARKET system*

To fully harmonize the electricity market system, there is a limited number of initial systemic assumptions one has to adopt:

- The electricity market system is vertically (and horizontally) structured; vertical decomposition follows the principle of vertically nested fractal-like subsystem systems with essentially the same characteristics ("cellular" systems). This applies to both segments, the market, and the grid.
- The complete system is top-down unbundled: roles and processes in the grid segment, which constitute public function, are separated from those in the market segment, which are commercial.
- The structure of the system must be such as to pursue the system-wide optimum within each subsystem level, taking into account the global objective of decarbonization. In practical terms, this applies to the MBA and lower.

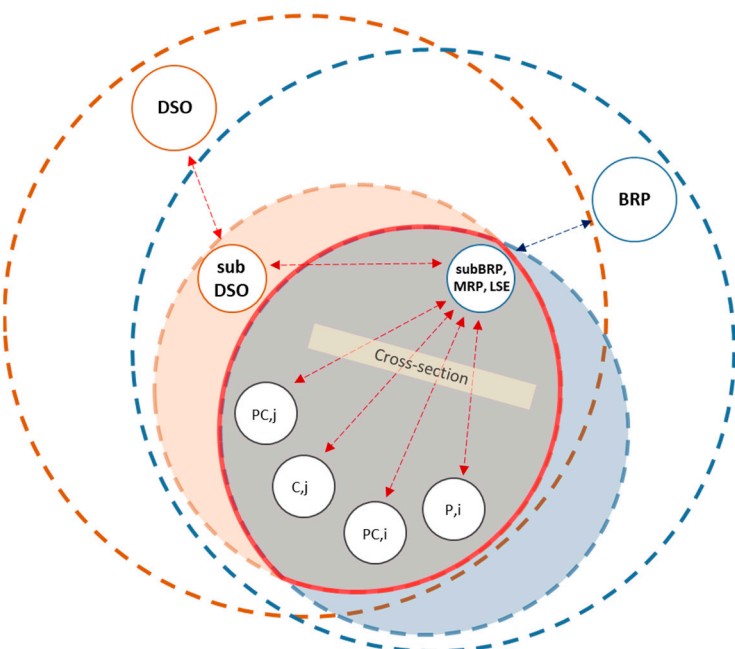

**Figure 8.** Cross-section between sub-DSO territory and (sub-BRP and micro-grid) or local energy community system.

*2.6. New Use Cases, Roles, and Business Models*

2.6.1. Use Cases as Scalable Segments of the Electricity Market System

The concept of use cases is used for several types of applications, typically for providing a testing and operating environment for the concrete function or functionality of a system. In harmonizing the electricity market system, the concept of use cases is applied on a systemic level. *Systemic use cases* represent the primary subsystems of the electricity (energy) market system. They are defined as "end-to-end", i.e., involving all processes and all players' roles in energy flexibility trading. Due to this, they have several important characteristics:

- The use cases cannot be defined to conflict with each other.
- There is a limited number of Systemic use cases within the electricity market system.
- They are scalable since the subsystems are "cellular"; this facilitates integrations in the market and the deployment of technologies.
- The use cases can be combined.
- The use cases can be connected.

Systemic use cases are instantiated in different environments, with different regulatory and system constraints, and with different role players and different business models. These instantiated use cases multiply the number of use cases but must be considered as "nested" sub-cases within applicable Systemic use cases.

GOLFEX termed Systemic use cases "primary". The project FEVER structured all the use cases into different categories: Instantiated Systemic Use cases were termed "High-level Use cases" (HLUC), as opposed to Primary Use cases (PUC) and Secondary use cases (SUC) which were defined for different functions [9].

2.6.2. New Roles

For further harmonization of the electricity market system, the new roles mostly address a unit process from the viewpoint of present technologies, while the existing roles mostly refer to what has now become an actor in the system and consist of more unit roles.

The former are termed *Technical roles* and the latter *Business roles*. To support convergence in the harmonization process and permit merging with the existing model, it is necessary to properly define and use both Business roles and Technical roles.

Business roles are the actors defined coherently with the adopted assumptions. A coherent definition implies that, within adopted assumptions, they do not conflict with other Business roles. They normally combine a certain number of Technical roles to represent a business system, with objectives and boundaries constrained by the boundary conditions of their environment.

Technical roles represent the basic level of the decomposition of Business roles, as made possible by the technologies available and as needed by the Business roles. They are, in fact, functions—unit processes and not systems. Typically, they can be envisaged as "services" enhancing the functionality of the (main) business role (actor) which spawned their development or services to be offered by the Business role to other Business roles.

The description of technical roles is out of the scope of this paper.

### 2.6.3. New Business Model for the DSO

Within the new use cases, a number of new business models will be generated by market players with large or substantial business potential. Of these, the new business model for the DSO is of particular importance.

The cellular approach to grid system structuring entails the following main constituents of the business model for the DSO:

- For the local balancing of energy flows on the distribution grid, it engages sufficient ancillary services that include the use of the energy flexibilities of the prosumers by the DSO for short-term reserves.
- To service time-critical transients and enable localized problem-solving for an enhanced observability system with short-term forecasting capability
- The new activities (long-term and short-term) necessitate the use of the avoided costs principle for business operation, applied also for the partitioning of funds from the network fee between the TSO and DSO. This will provide for an optimized investment policy for both TSOs and DSOs.
- The sufficient offering of localized energy flexibilities is enabled by the use of the dynamic prices of energy (flexibilities) based on the actual local conditions of the grid.

The description of other business models is out of the scope of this paper.

## 3. Evolution of the Electricity Market

As mentioned and referenced in Section 1.1, the authors of the HEMRM, ENTSO-E, ebIX, and EFET kept updating and modifying the HEMRM on a regular basis. However, in parallel to this process, the research community in Europe's first within Framework programs and H2020 Research and innovation projects generated a number of new concepts and solutions, based on the orientation of the SET plan and the Winter Package. In particular, the Innovation projects in H2020, which introduced pilot demonstration projects in a real environment, represented and continue to represent a concentrated scope of ideas. Since 2016, these have been tracked and jointly pursued within the BRIDGE community of projects. This process finally produced an initiative concerning updating the HEMRM.

### 3.1. Initiative within the Bridge Community of Projects

At the BRIDGE community General Assembly in 2020, the initiative was started to further harmonize the electricity market system in Europe, carrying the original HEMRM, shaped by ENTSO-E, EFET, and ebIX to new lower levels of the electricity system.

In 2020, a task force group was formed. In the first phase, it gathered the ideas and proposals from different concerned IA projects—"First selected projects" (participants from several concluded and ongoing projects participated: CoordiNET, EU-Sys-Flex, FEVER, GOFLEX, INTERRFACE, PlatOne).

Subsequently, in 2021 (first meeting 26.5.21, second meeting 1.07.21, and third meeting 23.9.21), the projects were discussed by a Joint Expert Group with the representatives of electricity system stakeholders—the three authors of the HEMRM ENTSO-E, EFET, and ebIX—with the inclusion of the four DSO organizations (CEDEC, EDSO for Smart Grids,

Eurelectric, and Geode). The final report was submitted to the EC for final approval in November 2021. Public versions were published by both BRIDGE and ENTSO-E [15].

Observation of the Present State of the Differential Analysis and Conclusions

The Group focused on roles: the Existing HEMRM roles (2020 issuance) were discussed, several updated descriptions—integrations on existing roles—were proposed, possible actors—players in actual electricity market systems—were listed, and 20 new or modified roles were proposed by Bridge projects. A few important ones to harmonize the system downwards include:

- Data Owner;
- Data User;
- DSO;
- Energy Transfer Cost Calculator;
- Flexibility Services Provider;
- Local Flexibility Market Operator;
- Regional Flexibility Market Operator for DSOs;
- Sub-Balance Responsible Party;
- Sub-DSO;
- TSO;
- TSO-DSO Coordination Platform Operator.

The discussions and meetings of the Group were observant and conducive to the convergence and continuation of the process. On the other hand, the participants started from different positions in the electricity market system and had understandably different starting viewpoints on the necessary evolution of the roles and process model. By implication, this would necessitate further work and several steps in reaching convergent views.

When comparing the existing roles with the newly proposed ones, it was clear that the new roles mostly address a unit process from the viewpoint of present technologies, while existing roles mostly refer to what has now become an actor in the system and consist of more unit roles. As defined in Section 2.6.2, the former are Technical and the latter Business roles. This distinction may support convergence in merging the new roles in the existing HEMRM.

An aspect of the HEMRM not addressed in the BRIDGE initiative and not discussed in the Joint Expert group, are "*domains*"; in HEMRM, they are subsystems of the Electricity market system and represent the structuring of the system, as described in Section 1.2. Consequently, there was no discussion of use cases, and there was no explicit discussion about the further structuring of the electricity market system, as proposed in Section 2.

### 3.2. Extension of the HEMRM and Evolution of the Electricity Supply System in Europe

The driving reason for the inception of HEMRM is the need for a standardized framework across the electricity supply system in Europe.

Based on this need, the HEMRM has represented, from its early formulation, a lighthouse for the evolution of the electricity supply system in Europe and has been followed up by European directives, which diffused with different rates into national regulatory systems and actual systems. While the process started at the turn of the millennium, the current state of energy supply systems in member states still ranges from vertically structured—harmonized to the level of the first formulation of HEMRM—to vertically integrated systems.

Consequently, the initiatives for the evolution of the system coming from research and innovation projects through the BRIDGE community are oriented toward the evolution of the HEMRM and not at concrete national regulations. Because of this process, the national regulations mostly represent constraints and barriers that hinder the introduction of new system concepts and limit the possibilities for testing them in real electricity supply system

environments in different states. The approach advocated and proposed by BRIDGE has been to create a pilot regulatory environment for demonstration cases.

This approach has also been used by the project GOFLEX and—where supported by a responsible demonstration case partner—resulted in a de facto "sandbox" regulatory environment that enabled implementation in a real environment. In some countries, this can constitute a bottom-up incentive for changing the national regulatory system, which can complement the systemic push provided by the upgraded HEMRM.

## 4. New Systemic Cases of the Electricity Market System

### 4.1. New Systemic Use Cases—Local and Regional Markets

As discussed in Section 2.6.1, the cellular structure of the harmonized system generates new systemic use cases in the extended part of the system. In line with the system structure, these can extend into regional and local markets, energy markets and markets for ancillary services for the DSO, and islanded and connected subsystems.

Two groups of use cases stem from the new business model for DSO and are enabled by harmonization (Section 2.6.3):

- Ancillary services of the DSO, including power reserves for balancing the local grid;
- A balancing market for the DSO, both local and regional (intra-MBA).

In this paper, the new systemic use cases defined by project GOFLEX and project FEVER are described in the following Section 4.2.

Of these, two use cases may be considered representative of the new markets:

- Connected Local Energy Community, cf. Figure 6 and the description in Section 2.
- Regional balancing Market for energy flexibilities for DSOs, cf. Figure 9 and description below.

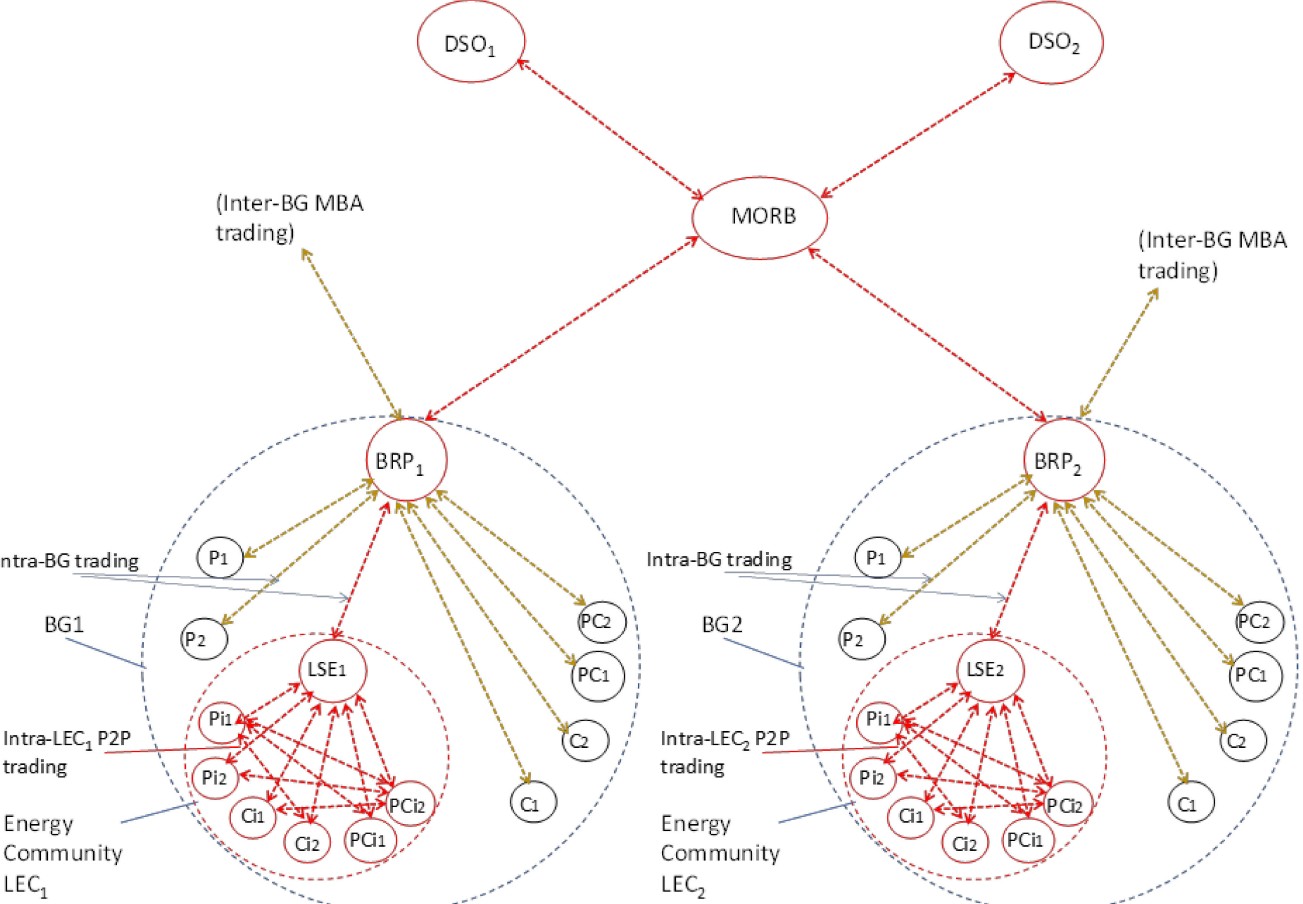

**Figure 9.** Regional balancing market for energy flexibilities for DSOs.

A regional balancing market for energy flexibilities for DSOs is established between participating LECs as providers and their DSOs as buyers of flexibilities. LECs participate with their available flexibilities through their parental BRPs. Regional trading is combined with intra-LEC trading (GOFLEX and UC6); if involved LECs establish the same system of values and objectives, a joint unique trading system for LEC members can be formed (FEVER and HLUC-14). To enable the joint optimum, the energy transport costs along the distribution grid are included.

For a description of the acronyms involved in MBA trading, cf. the legend in Figure 5 in Section 2.1; for description of new roles and their acronyms, cf. Tables 1 and 2 in the following Section 4.2.

**Table 1.** Overview of use cases enabled as part of the GOFLEX and FEVER role and process models.

| UC No. | Use Case | EM Subsystem | Business Role | Grid Operator | Grid Subsystem | Nesting Level | Trading In | Type of Trading |
|---|---|---|---|---|---|---|---|---|
| Local energy community and micro-grid | | | | | | | | |
| UC2 | Optimized operation of micro-grid | LCM | MRP | SDSO | subDistG | local | en.flex | 1:many |
| UC2-1 | Islanding operation of micro-grid | LCM | MRP | SDSO | subDistG | local | en.products | many:many |
| UC5 | Local energy community | LEC | LSE/ECR | SDSO | subDistG | local | en.flex | 1:many |
| UC5-1 | Islanding operation of local energy com | LEC | LSE | SDSO | subDistG | local | en.products | many:many |
| HLUC-5 | Flexibility exploitation for islanded micro-grid operation | LCM/LEC | MRP/ LSE | DSO/ SDSO | DistG/ subDistG | local | en.products | many:many |
| HLUC-8 | Economically optimized flexibility leveraging for a connected micro-grid | LC/LEC | MRP/ LSE | DSO/ SDSO | DistG/ subDistG | local | en.flex | many:many 1:many |
| HLUC-13 | Improving the outcome in flexibility by introducing sector coupling | LC/LEC | MRP/ LSE | DSO/ SDSO | DistG/ subDistG | local | en.flex | many:many 1:many |
| HLUC-15 | P2P flexibility trading | LEC | LSE | DSO/ SDSO | DistG/ subDistG | local | en.flex en.products | many:many 1:many |
| Balance Group | | | | | | | | |
| UC3A | Optimized operation of Sub-Balance Group | SBG | SRP | (TSO) | MBA | regional | en.flex | 1:many |
| UC3A-1 | Marketplace system for energy in BG (SBGs) | BG | BRP | (TSO) | BG | regional | en.products | many:many |
| UC3 | Optimized operation of Balance Group | BG | BRP | (TSO) | MBA | regional | en.flex | 1:many |
| UC3-1 | Marketplace system for Energy (BRPs) | MBA | BRP | (TSO) | MBA | regional | en.products | many:many |
| Ancillary services for DSO | | | | | | | | |
| UC4 | Congestion management at DSO | BG | BRP/FSP | DSO | DistG | local | en.flex | 1:many |
| UC4-1 | Local Balancing market for en.flex for DSO | BG | LMO | DSO | DistG | local | en.flex | 1:many |

**Table 1.** *Cont.*

| UC No. | Use Case | EM Subsystem | Business Role | Grid Operator | Grid Subsystem | Nesting Level | Trading In | Type of Trading |
|---|---|---|---|---|---|---|---|---|
| HLUC-2 | Leveraging the batteries' inverters towards reactive power ancillary services | BG | LMO | DSO | DistG | local | en.flex | 1:many |
| HLUC-12 | Creating dynamic tariffs based on flexibility use in the actual regulatory framework | BG | LMO | DSO | DistG | local | en.flex | 1:many |
| Regional Markets | | | | | | | | |
| UC6 | Regional Balancing Market for en.flex for DSOs | MBA | MORB | DSOs | DistG | regional | en.flex | many:many |
| UC7 | Regional Market for Energy Flexibilities (for BRPs) | MBA | MORF | TSO | MBA | regional | en. products | many:many |
| HLUC-14 | Form a first example of a regional flexibility exchange model | BG | MORB | DSOs | DistG | regional | en.flex | many:many |

**Table 2.** Legend for the acronyms used in Table 1.

| Acronym | Name |
|---|---|
| MBA | Market Balance Area |
| BG | Balance Group |
| LCM | Local Community Micro-grid |
| LEC | Local Energy Community |
| SGB | Sub-Balance Group (Balance sub-group) |
| MO | Market Operator |
| BRP | Balance Responsible Party |
| FSP | Flexibility Service Provider |
| SRP | Sub-balance Group Responsible Party |
| MRP | Micro-grid Responsible Party |
| LSE | Local Supplier of Energy |
| LMO | Local Market Operator |
| DSO | Distribution System Operator |
| SDSO | Sub-DSO |
| MORB | Market Operator for Regional Balancing Market for DSOs |
| MORF | Market Operator for Regional Market for Energy Flexibilities |
| TransG | Transmission Grid of TSO in MBA |
| DistG | Distribution Grid of a DSO in MBA |
| subDistG | Sub-Distribution Grid of a DSO, belonging to SDSO |
| FSP | Flexibility Service Provider |
| ECR | Energy Community Responsible |

*4.2. Short List of Possible New Use Cases—Local and Regional Markets*

*GOFLEX Use cases*. The GOFLEX use cases are those use cases in the MBA that are enabled by GOFLEX integrated solution using the GOFLEX roles and process model. They comprise:

- The use cases within the Harmonized Electricity role model;
- The new use cases made possible through further vertical structuring of the electricity market and harmonization of joint and supportive processes—the electricity grid system.

The GOFLEX project focus is local, with DSO as the dominant user of energy flexibility for avoiding congestion and achieving local balancing of the grid. Only use cases (UC) involving the trading of local subsystems are listed.

*FEVER Use cases.* The FEVER project addresses both local, regional, and established markets in the MBA. The high-level Business use cases (HLUC) addressing systemic (primary) use cases are presented. Only those HLUCs describing the complete segment of the system are listed.

A combined summary overview of possible new use cases for local and regional mar-kets is given in Table 1 below; the acronyms of the terms used are explained in Table 2.

### 4.3. Roles in New Use Cases

In line with the use cases, the roles used in the roles and process model are:

- Roles of the Harmonized Electricity Market Role Model;
- New roles due to its cellular extension of the electricity market segment downwards;
- New roles due to the proposed cellular structuring of the electricity grid segment.

Additionally, Technical roles are further structured to accommodate the requirements of the currently available and used technologies; they differ from project to project. The extent and the rate of convergence depend on the extent and the rate of the adoption of harmonization in the system (cf. Section 3). A facilitator of this process is the standardized flexibility trading protocol, FlexOffer [5].

The roles, in particular the new Technical roles, can be integrated to suit the business models of actual players in different use cases.

In Table 1, only Business roles are listed. Technical roles, defined by GOFLEX and FEVER, are not listed, as they are not technology agnostic. For both systems and roles, the GOFLEX terminology for roles–actors, is used to simplify comparison. However, the new FEVER roles not identified in GOFLEX are listed in parallel, if considered to be the driving use case roles.

The GOFLEX roles and process model [7] include only roles participating in the GOFLEX trading processes; the interacting and other roles are the roles of the Harmonized Electricity role model. The main roles driving the new use cases in GOFLEX and FEVER are included in Table 1. The acronyms are explained in Table 2. The list of roles in GOFLEX can be found in [7], and the roles used in FEVER in [16].

## 5. Smart Harmonized Electricity Market System

Project GOFLEX identified the main characteristics of the Smart Harmonized Electricity market system and termed them GOFLEX technology enablers [7] (p.51). Accordingly, the Smart Harmonized Electricity market system is the system characterized by:

- A harmonized structure of the complete system—with vertically nested "cellular" systems;
- Automated trading of direct or aggregated energy flexibilities and flexible energy products, with trading intervals equal to tariff intervals or closer to real-time for critical transients;
- Energy flexibilities offered on the market by the prosumers, active consumers, and producers (collectively termed "Prosumers"). The flexibilities are extracted from energy reservoirs, either virtual or explicit;
- Flexibilities offered by prosumers that are purchased by other prosumers (peer-to-peer), traders, or system operators for ancillary services;
- Dynamic pricing established by forecasting the needs of the system operators for congestion avoidance and for balancing the grid;

- Enhanced observability and forecasting of local conditions on the grid;
- The use of the concept of net income/revenue based on the avoided cost principle to select the flex energy supplier and for the TSO–DSO partitioning of responsibilities and remuneration;
- Offering flexibilities, aggregating them, and trading, using one of the standardized open protocols, such as the generalized FlexOffer [5];
- Maximizing the available DR of the prosumers on the market; dynamic pricing based on local conditions on the grid is used;
- Energy transfer costs that are added to the costs of energy flexibilities when comparing the flex offers in the MBA (implicit transfer capacity trading). This enables overall MBA system optimization. The point of comparison is the topological location of (predicted) congestion or disbalance;
- Different users (TSO, DSO, BRP, etc.) competing for the same flexibility when using open market principles.

  Complete harmonization represents:

- Interoperability and scalability, which is achieved much easier in a harmonized system (=structured into vertically nested "cellular" systems);
  - A limited number of use cases (=markets);
  - The same (type of) roles (=unit actors);
  - The same (type of) unit processes.
- The synergy between system operators and prosumers; dynamic pricing based on local conditions of the grid.

### 5.1. Automated Trading of Flexibilities

Enabling technology for full harmonization of the energy system is the automated trading of energy flexibility and flexible energy products close to real-time and offered by the prosumers—all types of parties connected to the grid.

The trading intervals are, as a default, equal to tariff intervals but can be shorter, depending on the request of the buyer. Requests are based on the forecasted needs of the buyer. For time-critical events on the grid, trading intervals of 1 min are possible.

Automated trading enables a localized approach; thus, offers issued by prosumers closer to the point of predicted transient are prioritized, using the implicit trading of energy transfer capacity, as described earlier in this section.

Depending on the Systemic use case—the type of the market—automated trading can be a one-sided pool type (1: many) or a two-sided pool (many: many); e.g., in internal LEC trading, trading is many: many (peer-to-peer), cf. Table 1 in Section 4.2.

A complementary enabler to automated trading is an advanced distribution grid observability system that is capable of the localized close-to-real-time forecasting of the state of the grid.

### 5.2. Prosumers and Dispersed Energy Production

Bringing prosumers to the status of active participants in flexibility trading is closely connected with the creation of local energy communities and with dispersed energy production as opposed to concentrated energy production. This reduces the share of the transfer of large amounts of energy over large distances at transmission levels and increases the level of local/regional self-supply. Self-supply will then do away with or alleviate the impact of a single failure criterion of active components in the electricity grid.

With this second pillar, the paradigm changes from the "cascade/waterfall" transmission distribution of electricity to multi-path, two-directional flows of energy based on two complementary production sources. Two-directional energy flows significantly increase the availability of the energy supply and the energy produced in local subsystems will represent a second energy supply source. This is becoming a progressively more and more important aspect in the view of the security of supply.

*5.3. System-Wide Optimum in Smart Harmonized Electricity Market System*

Harmonization of the energy supply system enables the formulation of a system-wide optimum over a complete MBA system. The necessary prerequisite is that (i) the flexibility offers can be compared on an equal footing and (ii) in comparison, all the avoided costs of alternative solutions are taken into account.

Formal conditions, formulated in project GOFLEX:

- The location of the supply of flexibility and the topological location of the transient of the grid needing flexibility to avoid congestion and provide a balancing of energy flows on the grid;
- The calculation of flexible energy transfer costs between the location of supply and the topological location of the transient. The costs include both marginal investment costs and associated operational costs;
- The inclusion of calculated energy transfer costs, when comparing different offers, as part of the trading algorithm, i.e., including them as the implicit trading of energy transfer costs.

Three types of locations were defined:

- The geographical location of the party connected to the grid offering flexibility;
- Logical location: the calculated weighted location of the aggregated offer of several parties connected to the grid;
- Topological location: the calculated location of the predicted transient of the grid needing flexibility.

The concept has been applied and its validity tested both in flexibility trading and in investment decisions in two innovation projects: in project FEVER, in use case HLUC-14, (cf. Table 2) in setting up a first model of a regional flexibility exchange between two local energy communities [16,17], and in project GIFT [8], to compare techno-economic feasibility and select the supply of flexibility to the island grid from two possible subsystems.

*5.4. Energy Reservoirs and Techno-Economic Balancing*

An important property of functional systems is that they contain processes that enable the formulation of joint objectives and techno-economic optimums.

As noted in Section 2.6, vertically nested primary systems contain all processes and roles that participate in energy flexibility trading. Consequently, joint objectives of the system can be formulated, and the techno-economic optimum can be defined and set forth as a system design target to achieve a dynamic balance between the consumption and production of energy. To do this, both or one of them must be dynamically adapted. To enable the techno-economic analysis of investment decisions, project GOFLEX adopted and further defined the concept and the category of energy reservoirs. The energy reservoirs can be implemented in two ways: (i) by changing the dynamics of an existing production or ambiental process from which flexibility is extracted, this type has been referred to as an *implicit energy reservoir*, and (ii) by using dedicated systems for storing energy, referred to as *explicit energy reservoirs*.

The storing capacity of implicit energy reservoirs depends on the potential of the process for adapting the dynamics; the associated incurred costs compared with the revenues from the flexibility offered on the market are the basis of techno-economic analysis and determining criteria limiting the amount of flexibility offered.

Explicit energy reservoirs range from water reservoirs to different energy storage media. The types include batteries and different chemical energy storage systems. The costs incurred are both CAPEX and OPEX costs.

Implicit energy reservoirs are typically short-term, providing flexibility in the range from minutes to hours. Explicit reservoirs target and enable longer-term storage. Of these, hydrogen is emerging as a promising energy vector that will enable longer-term storage, including seasonal storage (cf. Section 6).

A significant characteristic of implicit energy reservoirs is that they reside in existing primary processes and typically require investment in only an (improved) energy management system; consequently, they represent the backbone of the demand response contribution to the available flexibility of the prosumers on the market. On the other hand, as their capacity and availability are limited by the requirements of the primary process, their overall techno-economic yield significantly depends on the good coupling of the process and energy management systems.

The use of implicit energy reservoirs represents an economic add-on to the primary process of the prosumer. Conversely, explicit energy reservoirs are process systems, entailing significant CAPEX investments and, in principle, the primary objective of storing energy. Consequently, they may also act as a business actor in the energy supply system.

The use of implicit energy reservoirs in flexibility trading has been tested and validated for both ambiental and production processes for a number of cases and several types of industries in GOFLEX, FEVER, and other innovation projects, with a predominant focus on the existing technologies of primary processes. With the new framework goal of achieving carbon neutrality, this segment has started a new technology cycle in which primary processes, in particular those in process energy-intensive industries, have to achieve carbon-neutral production by replacing fossil fuels with renewable fuels. This objective will generate new synergy with the hydrogen systems in the energy supply system, as discussed in Section 6. For concrete use case validation, please refer to Section 7.5.

## 6. Role of Bottom-Up Systems in Dispersed RES Supply

The LEC consists of prosumers, consumers, and producers of different sizes and characters: residential homes, tertiary buildings, RES producers, industrial companies of various sizes, and technology—all of which are connected to the electricity grid.

The local energy community (LEC) is one of the most relevant ecosystems in terms of green transition. It brings into the energy supply system new players—prosumers—who actively trade their flexibilities among themselves and on the external markets, and stimulates new enabling technologies, notably, automated close-to-real-time trading, boosting end-to-end automated solutions.

With the advent of hydrogen-based systems in energy supply, the LEC represents an energy supply subsystem that, on the one hand, boosts the local production of renewables, and on the other hand, links different energy carriers, thus providing a cross-sector optimum.

On-site integrated hydrogen-based systems connected to the grid, consisting of an electrolyzer, hydrogen storage, and fuel cell system—termed *hydrogen prosumers*—provide efficient balancing of the local energy consumption and local production of renewable energy that can be extended over annual cycles. There is no transport of hydrogen needed. Thus, *H2LEC*—a local energy community with hydrogen prosumers [18]—represents the carrier of dispersed energy (and hydrogen) production as a complement of concentrated energy and hydrogen production. On average, H2LEC will achieve at least 75% self-supply. Additionally, with combined heat-and-power systems, coupling to a thermal system (see Section 7) adds to energy efficiency. The authors forecast that this will influence the future evolution of the energy system to a more balanced repartition between the concentrated energy production in large power plants and dispersed energy production.

Additionally, it represents a virtual socio-economic system based on and sharing community values and objectives. In the context of a green transition, this leads to:

- Engaging the initiative, innovation, and capital of local actors; in particular, new technology start-ups and young generations.
- Support for including the cost of the degradation of the environment in the total CAPEX and OPEX functions.

The H2LEC also creates the need and the market for smaller integrated hydrogen-based systems ranging from a few kilowatts (kWe) for residential homes to a few megawatts (MWe) units for larger industrial companies, with a competing range of sizes in between

for public and tertiary buildings and smaller enterprises. Thus, they provide an opening for the participation of SMEs in international value chains and predictively create a strong complementary bottom-up pillar in the energy and hydrogen supply system in Europe.

The solutions and systems for H2LEC are in the innovation cycle and are appearing in the actual environment; the H2LEC are in the making. However, they face several hurdles and challenges; these include regulations and resistance to change by the vertically integrated energy sector and financing, in particular in the innovation cycle and early operation phase.

## 7. Discussion

### 7.1. Depassing the Monetary Value of Flexibility: Multidimensional Cost and Price Structure by Assets

In flexibility trading within local energy communities and other similar cellular subsystems, between members of the community, the cost structure of the flexibility can be extended from a purely monetary value, with reference to tariff prices, to include other categories, derived from joint community values and objectives. Depending on the business model of the energy community, this multidimensional cost structure may include joint investments into green infrastructure and the support of various community services. An attempt to introduce this multidimensional cost structure was defined in FEVER, introducing "pseudo currency" as a common value denominator [16] (pp. 227–235).

This approach can be shared by different LECs in inter-LEC trading on the regional level if they adopt the same community objectives and targets. A prerequisite is the introduction of the energy transfer cost calculation into the cost/price structure of traded flexibilities.

### 7.2. Future Extension to "Total Costs of Life" of a System (Including Environmental Costs)

The avoided cost principle can be extended to include other categories, besides monetary, to include environmentally avoided costs, introducing the concept of the "total cost of life" of a system into the energy supply and consumption system. With the shared objective of carbon neutrality between subsystems in the energy supply over a larger territory, this will predictably happen in phases. The first phase is the wide adoption of dynamic pricing based on the local conditions of the grid and an energy transfer cost calculation as part of the flexibility price in the distribution grid. The second phase will go hand-in-hand with the introduction of this concept into the prices of products and services on the market.

### 7.3. Sector Coupling, Multi-Vector Optimization

One important characteristic of the HEMRM is the coupled ("not unbundled") operation at the lowest levels of the system (cf. Section 2.4). This is intrinsically true at the level of the party connected to the grid/prosumer. Such entities are connected to several energy systems, such as electricity, heat, and gas, each representing its own energy vector. The techno-economic optimum at this level is a multi-dimensional problem, requiring complex tools and knowledge for holistic optimization.

A similar approach as within the prosumer can be applied to a local energy community (LEC), provided that the LEC is capable of local renewable energy production and that the infrastructure for multi-vector distribution is available. Such a community can optimize itself and its members on a local level, reducing energy transport requirements while providing sufficient multi-vector energy to the local population.

7.3.1. Typical Cases of Sector Coupling

Electricity and Thermal Distribution System

This use case can be seen as an extension of the UC-5, described in Section 4.2, and is also identified in FEVER as HLUC-13 [9] (pp. 59–61). The system consists of local RES production (e.g., PV), an electricity distribution system, sector-coupled units (e.g., biogas

CHP), a thermal distribution system, and end-users/prosumers connected to either or both energy networks. An example of such coupling is presented in Figure 10.

**Figure 10.** Example of electricity and thermal sector coupling with different types of producers, consumers, and prosumers.

The optimization approach is the following:

- Extend the description of flexibility by introducing a second dimension to the sector-coupled units and users.
- Assess the cross-sector interaction (diffusion) between the sector-coupled units and use the structured information in the optimization algorithm.
- Define the objectives of the LEC and prosumers in several dimensions.
- Couple the dynamics of both systems.

The last item is particularly important in the electricity–heat coupling, as the time dynamics of the two systems differ by a large margin [19]. For example, a change in the power output of a CHP has an almost immediate effect on the electricity network; however, the effect on the thermal network will be evident much later, depending on the structure of the system, energy carrier type, and other parameters. Such rebound effects must be considered in the optimization algorithm and are a key to local optimization and multi-vector self-supply.

Electricity and Transportation (EV)

This extension can be applied to any of the use cases described in Section 4.2 and can also be an additional extension of the electricity-thermal coupled use case. Local use cases are preferred due to the reduced need for the transport of electricity for EV charging over large distances.

The approach to optimization is the following:

- Extend the description of flexibility by introducing a second and/or third dimension to the sector-coupled units and users.
- Automatically assess the dynamics of the particular charging process by using input parameters, technical parameters, analyzed user behavior, and big-data insights.
- Based on the assessment above, exploit the flexible portion of the charging process with a high degree of probability.

As with thermal coupling, transport coupling introduces additional constraints and optimization dimensions. The key parameter is the (un)availability of the resource—EV—which shall be assessed automatically and with high reliability.

7.3.2. Holistic System-Wide Optimum

The energy system is dynamic and interdependent across different vectors and networks. Managing/optimizing only a portion of the system ends in local optima, which can conflict with global optimum. This holistic view is not available on the local level and can only be accessed through an efficient systemic structure based on vertically nested subsystems. The holistic optimization of the system (with all of its children) is harmonized and coordinated through the automated process of flexibility exchange within and outside the particular level.

Considering the sector-coupled nature of the system at the lower levels of optimization and actual steering brings the following benefits:

- Economic gains for the sector-coupled entities;
- Less energy conversion (and therefore losses);
- Less energy transportation (and therefore investments and losses);
- Intrinsic consideration of multi-dimensional constraints.

The last item enables the exploitation of bottom-up flexibilities as it avoids congestion and other constraints in several energy sectors. This information is not available in existing flexibility schemes (e.g., ancillary services for TSO) where only the (active) electricity component is considered.

*7.4. Impact of Vertically Nested Subsystems on Strategic Planning and Policy*

The assertion of vertically nested subsystems like LEC and H2LEC in energy supply systems in Europe, in conjunction with hydrogen-based systems, will significantly impact the planning on the member state level.

Integrated hydrogen-based systems will be the necessary building block in a sustainable and carbon-neutral economy of the future, particularly by complementing the concentrated production of big baseload power plants and renewable energy fields as well as reducing costs and risks of energy transfers over large distances. As part of local energy ecosystems, hydrogen not only enables local energy communities but also provides for their optimization as vertically nested subsystems in energy supply. In this way, new hydrogen technologies pave the way to the creation of new business models and stimulate the appearance of new market players on the level of energy communities and prosumers.

The authors estimate that the level of self-supply in H2LEC will be on average 75%. The rate of implementation will depend on further support of this approach within EU innovation projects and on the resulting evolution of the HEMRM.

This topic has been addressed in some other works by the members of the team; in particular, within the Temporary Working Group on Green Hydrogen, as part of the activities for upgrading the SET plan. However, this is not part of the reported work and is not quotable yet.

*7.5. Implementation of Harmonization Concepts*

The concepts for the further harmonization of the energy system presented in this paper have been the basis on which the business models for the three demonstration sites with three different use cases in project GOFLEX have been designed: for a university campus and dispersed prosumers in Cyprus [20], for the ESR for the Swiss pilot [21], and for the SWW in Germany [22]. They were all implemented and demonstrated within the project and after it in the evolution of their use cases. The business model of the SWW, the so-called "Four-step roadmap to the future", has been further carried out after the completion of the GOFLEX project, within FEVER and other innovation and investment projects. Presently, the LEC SWW includes sector coupling and hydrogen-based systems and is shaping up as a "Local Energy community as a Small Hydrogen valley" [18].

The concept proposed is already used in actual operations within the active Redispatch 2.0 system for grid state estimation, including forecast and virtual and explicit storage facilities, and in tracing and implementing further steps in creating, building, and operating an Energy Sharing Community within the existing legal and regulatory framework in Germany, as defined and furthered by the national energy agency [23].

## 8. Conclusions

Full harmonization of the electricity market system represents the systemic standardization of roles and processes, unique generic description of Systemic use cases, nonconflicting combining of markets, and easy scalability of solutions. In terms of strategic impact, it positions vertically nested subsystems such as the LEC and H2LEC as subsystems in energy supply, which enables both their optimized operation and system-wide optimum and brings them into the energy supply system as active traders and new business actors—involving subsystems and prosumers of all types, including residential.

The authors believe that the approach has the potential to create systemic synergy to boost the next post-COP cycle and will represent a change in the paradigm of energy supply. The dissemination potential within the GOFLEX project where this concept originated was limited both in scope and in time and to realize its full impact, it should be given the support of the BRIDGE instrument to enable its further diffusion into HEMRM and, through it, into the actual energy system of Europe.

Furthermore, the process of European market-oriented structuring and integration based on full harmonization can and will contribute substantially to the backbone of the "European way" on a global scale. Characterized by the coherent interweaving of new technologies, new business models, and new business actors, it has the potential to play a significant role in reforming electricity markets in other countries and regions as the first target and, in particular, in Mission Innovation countries.

**Author Contributions:** Harmonization concept of vertically nested systems, Z.M.; cross-sector coupling and multi vector optimization, S.B. and Z.M.; use case and roles definition, Z.M. and S.B.; exploitation of harmonization concepts and use case implementation, G.M.; writing—original draft preparation, Z.M. and S.B.; writing—review and editing, S.B. and G.M. All authors have read and agreed to the published version of the manuscript.

**Funding:** The projects MIRABEL, GOFLEX, GIFT, and FEVER have received funding from the European Commission's research and innovation program: project MIRABEL under Grant Agreement no. 248195 (FP7), project GOFLEX under Grant Agreement no. 731232 (H2020), project GIFT under Grant Agreement no. 824410 (H2020), and project FEVER under Grant Agreement no. 864537 (H2020).

**Data Availability Statement:** All the referenced public documents are available on request from the authors. The data presented in this paper that are not publicly accessible are available on request from the corresponding author.

**Acknowledgments:** The authors gratefully acknowledge the contributions of other Mirabel, GOFLEX, FEVER, and GIFT projects' authors and team members for their contributions to the work reported.

**Conflicts of Interest:** The funders had no role in the design of the study, in the collection, analyses, or interpretation of data, in the writing of the manuscript, or in the decision to publish the results.

## List of Abbreviations and Acronyms Used in the Paper

Note: the names of roles are not included. The acronyms of roles in new systemic use cases are explained in Table 2 in Section 4.2.

| | |
|---|---|
| BA | Balance Area |
| BG | Balance Group |
| CHP | Combined Heat and Power |
| ebIX | European Forum for Energy Business Information Exchange |
| EFET | European Federation of Energy Traders |
| ETSO | European Transmission System Operators (presently ENTSO-E) |
| EUW | European Utility Week |

| HEMRM | Harmonized Electricity Market Role Model (abbreviated also: Harmonized Roles model or Harmonized Electricity Roles model, Harmonized model) |
|---|---|
| H2LEC | Local Energy Community as a Small Hydrogen Valley |
| HLUC | High-Level Use Case |
| LEC | Local Energy Community |
| MBA | Market Balance Area |
| PUC | Primary Use Case |
| RES | Renewable Energy Source |
| SGB | Sub-Balance Group |
| SUC | Secondary Use Case |
| UC | Use Case |

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
