# Peer review of "Extension of the HEMRM—Full Harmonization of the Electricity Supply System"

_electricity, doi:10.3390/electricity5010003_

Round 1
Reviewer 1 Report
Comments and Suggestions for Authors
This paper introduces the full harmonization of the electricity supply system in Europe. The topic is interesting and meaningful, and I have some concerns to improve the quality of the paper:
1. The HEMRM appearing in the abstract should also be given its full name.
2. It is necessary to add a table to summarize the differences between the European market before and after the Extension of the HEMRM.
3. It is recommended to provide an overall content introduction at the end of the first chapter of the article to enhance the readability of the article.
4. Can the process of European market-oriented integration have reference significance for the reform of electricity markets in other countries and regions? It is suggested to supplement in the discussion.
5. What is the fundamental reason for the full harmonization of the electricity supply system in Europe, and the impact of this process on overall market efficiency and energy transformation should also be highlighted.
6. The article involves a large number of abbreviations. Although the author has provided the meanings of the relevant abbreviations in Table 1, it is recommended to provide the full names corresponding to all abbreviations at the beginning of the report to enhance the readability of the article.
Comments on the Quality of English LanguageMinor editing of English language required
Author Response
uploaded: Authors Reply to the Review Report (Reviewer 1).pdf

Reviewer 2 Report
Comments and Suggestions for Authors
This paper presents a coherent approach to extension of HEMRM to the lowest levels in both grid and market segments – full harmonization. In my opinion, this is a well structured and written review paper. My suggestion to the editors is to accept the paper in present form.
Comments on the Quality of English LanguageModerate editing of English language is required.
Author Response
No response is necessary.
Reviewer 3 Report
Comments and Suggestions for Authors
1. Systemic Standardization and Scalability: The approach for full harmonization emphasizes the need for a standardized framework across the electricity supply system. This standardization would facilitate scalability, but it's crucial to consider how this framework will accommodate future technological advancements and regulatory changes.
2. Integration of Prosumers and New Business Actors: The inclusion of prosumers and new business actors into the energy supply system is innovative. However, it's important to assess how their integration will be managed in terms of grid stability, regulatory compliance, and market dynamics.
3. Automated Trading of Flexibilities: The proposal for automated trading of flexibilities by prosumers is forward-thinking. It would be beneficial to delve deeper into the required technological infrastructure, data security measures, and the algorithmic complexity of such a trading system.
4. Energy Reservoirs and Technoeconomic Balancing: The role of energy reservoirs, both implicit and explicit, in technoeconomic balancing is crucial. A deeper analysis of their capacity, efficiency, and integration into the grid for both short-term and long-term storage solutions would be insightful.
5. Coupling at Lower Levels and Cross-Sector Integration: The concept of coupling at lower levels and its intersection with other energy vectors is intriguing. It raises questions about the technical challenges and opportunities in cross-sector energy integration, especially in terms of grid management and energy efficiency.
6. Vertical Nesting and System of Systems Approach: The vertical nesting and system of systems approach suggest a hierarchical yet interconnected framework. Further elaboration on how this structure would manage dependencies and interactions between different subsystems would be valuable.
7. Impact on Strategic Planning and Policy: The proposed model's strategic impact, especially in positioning vertically nested subsystems like LEC and H2LEC, requires a deeper exploration of its implications for long-term energy policy and planning at both national and European levels.
8. Potential for System-Wide Optimization: The claim of system-wide optimization through this approach is significant. It would be beneficial to see a more detailed analysis of how this optimization is quantified and the metrics used to evaluate its effectiveness.
9. Role of Implicit Trading in Grid Stability: The concept of implicit trading of energy transfer capacities along the distribution grids is innovative. A detailed examination of how this trading would function in real-time and its implications for grid stability and energy pricing would be crucial.
10. Challenges in Implementation and Dissemination: While the approach has potential, the challenges in its implementation and dissemination, especially in the context of the GOFLEX project and the BRIDGE instrument, should be critically analyzed. This includes looking at technological, regulatory, and market barriers that might impede its broader adoption.
Comments on the Quality of English LanguageExtensive editing of English language required
Author Response
uploaded:
Author's Reply to the Review Report (Reviewr 3)

Round 2
Reviewer 3 Report
Comments and Suggestions for Authors
Unfortunately, comments 4 and 8 have not been adequately addressed. I strongly recommend incorporating at least two additional case scenarios from diverse perspectives to validate the practicality of the proposed concept in this study.
Comments on the Quality of English LanguageModerate editing of English language required
